# A multiple instance learning approach for detecting COVID-19 in peripheral blood smears

**Colin L. Cooke**[1], **Kanghyun Kim**[2], **Shiqi Xu**[2], **Amey Chaware**[2], **Xing Yao**[2], **Xi Yang**[2], **Jadee Neff**[3], **Patricia Pittman**[3], **Chad McCall**[3], **Carolyn Glass**[3], **Xiaoyin Sara Jiang**[3], **Roarke Horstmeyer**[1,2] *

**1** Electrical and Computer Engineering Department, Duke University, United States of America, **2** Biomedical Engineering Department, Duke University, United States of America, **3** Department of Pathology, Duke University Medical Center, United States of America

* rwh4@duke.edu

**Data Availability Statement:** Due to the conditions and agreements under which this data was collected, no data will be made publicly available, but can be requested at https://irb.duhs.duke.edu/ under IRB Pro00105472-KSP-2.0 - Cell

## Abstract

A wide variety of diseases are commonly diagnosed via the visual examination of cell morphology within a peripheral blood smear. For certain diseases, such as COVID-19, morphological impact across the multitude of blood cell types is still poorly understood. In this paper, we present a multiple instance learning-based approach to aggregate high-resolution morphological information across many blood cells and cell types to automatically diagnose disease at a per-patient level. We integrated image and diagnostic information from across 236 patients to demonstrate not only that there is a significant link between blood and a patient's COVID-19 infection status, but also that novel machine learning approaches offer a powerful and scalable means to analyze peripheral blood smears. Our results both backup and enhance hematological findings relating blood cell morphology to COVID-19, and offer a high diagnostic efficacy; with a 79% accuracy and a ROC-AUC of 0.90.

## Author summary

In this paper, we present a multiple instance learning-based approach to aggregate high-resolution morphological information across many blood cells and cell types to automatically diagnose COVID-19 at a per-patient level. We integrated image and diagnostic information from 236 patients to demonstrate not only that there is a significant link between blood and a patient's COVID-19 infection status, but also that novel machine learning approaches offer a powerful and scalable means to analyze peripheral blood smears. Our results both backup and enhance hematological findings relating blood cell morphology to COVID-19, and offer high diagnostic accuracy. Besides the final aggregated decision, the proposed attention mechanism also provides cell-type importance, which can help pathologists to build valuable insights on which cell types are more diagnostically relevant, opening a window into improving the explainability of deep optical blood analysis approaches.

morphology of COVID-19-positive blood smear images. The code used within this work has been made publicly available at: https://github.com/clvcooke/covid-blood.

**Funding:** This study was funded by a Duke-Coulter Translational Partnership, a fellowship from the Natural Sciences and Engineering Research Council (NSERC) of Canada, and funding from a 3M Nontenured Faculty Award. The funders had no role in study design, data collection and analysis, decision to publish, or preparation of the manuscript.

**Competing interests:** I have read the journal's policy and the authors of this manuscript have the following competing interests: RH is the scientific director of Ramona Optics Inc., and RH and AC are co-founders of Airilabs LLC. Both companies are developing novel hardware for microscope imaging.

# 1 Introduction

The analysis of blood cell morphology plays a critical role in hematology to diagnose and understand various diseases [1]. A key tool for blood cell morphology assessment is the light microscope, which is often applied to examine peripheral blood smears (PBS) [2]. In a typical procedure, a physician will visually examine white and red blood cells within a PBS on a glass slide at high microscope magnification (usually 100×). The nature of visual examination at high resolution limits the observable field-of-view (FOV) to contain just a few white and red blood cells at a time, making analysis of multiple cells challenging and time consuming. Digital microscopes [3] have emerged as an effective alternative to manual analysis. By automating the scanning process and presenting digitized images of PBSs to physicians on a computer, such digital microscopes are quickly becoming the predominate method of PBS analysis.

The digitization of PBS imagery has also led to new opportunities to apply advanced machine learning algorithms, such as deep learning methods, to examine blood data [4–7]. However, thus far, most algorithmic methods have relied on preexisting and developed under-standings of the morphological features of interest, to either facilitate the design of feature extraction techniques or to specify per-cell labeling criteria. This constraint limits the application of machine learning to automate the decisions that physicians are currently able to make, rather than providing fundamentally new capabilities or insights.

The limitations of current machine learning methods for blood cell morphology analysis were recently highlighted during the COVID-19 pandemic. There is a growing body of evidence that suggests COVID-19 and blood have a complex set of interactions that lead to significant morbidity and mortality [8, 9], and there is a variety of clinically reported evidence that COVID-19 induces morphological changes to both white and red blood cells [10–13]. However, our current limited understanding and agreement regarding such morphological impact have impeded development of effective blood-based diagnostic and prognostic screening tools.

In this work, we argue that a new way of analyzing blood, which we term *Deep Optical Blood Analysis* (DOBA), circumvents the need to pre-define features of interest or label individual cells within particular categories. It instead allows for an entirely data-driven analysis of blood using only patient-level information. DOBA uses deep learning to develop a mapping between images from a patient's PBS and their condition. In a typical digital PBS scan, images of hundreds of white and red blood cells are captured per-patient. It is therefore desirable to examine each image in detail, without requiring labels on the individual images. To accomplish this, we adopted a *Multiple Instance Learning* (MIL) [14] technique to link a patient's COVID-19 diagnosis (obtained with a standard PCR-RT laboratory test) to their blood image data. Specifically, a recent approach [15] paired an MIL attention mechanism with a convolutional neural network to simultaneously learn how to extract information from individual images and aggregate information across multiple images. We extended this work to form a novel hybrid MIL network based upon *model ensembling* [16], and applied our new algorithm to produce accurate final per-patient screening results directly from blood image data.

We chose COVID-19 as a case-study in developing DOBA due not only to the significance of the disease, but also the growing medical consensus regarding its connections to blood [17]. Despite this consensus, there is no convergence on a particular expression of COVID-19 in blood, with responses ranging from Thrombocytopenia [18], to COVID-19 induced blood clots [19], to morphological abnormalities [12, 13]. Therefore, despite sufficient evidence to attempt to use blood cell morphology to detect COVID-19, there is no clear starting point in examining individual cells using standard supervised machine learning approaches (i.e. labeling each cell individually).

Our new method not only enables diagnosis of COVID-19 without requiring such labeling, but also sheds light on how this new disease affects blood, by automatically producing a statistical summary of which specific cells and cell types are more or less important for the COVID-19 diagnostic task. Further, by applying specific *perturbations* to our image datasets, we have also developed a procedure to highlight which spatial features of the acquired image data were more or less important to enable robust screening. Apart from enhancing our understanding of the disease, these features also offer a window into algorithm operation to improve the explainability and reliability of our approach. We are hopeful that our new learning-based data aggregation strategy can serve as a starting point for future algorithmic strategies to elucidate the hematological impact of COVID-19 and other blood-related diseases.

## 2 Results and discussion

### 2.1 Study design

We investigated the diagnostic potential of PBS images for COVID-19 infection through a partnership with the Duke University Medical Center. Over a five-month period (April 2020 —August 2020) we collected digital PBS image data from 236 patients, 53% of whom tested positive for COVID-19 by a separately administered PCR test. We denote this group as the *Standard* cohort. No other patient information was collected for this cohort. In addition to the Standard cohort, we collected PBS image data from 40 additional patients admitted to the medical intensive care unit who presented with acute respiratory illness but were confirmed to be COVID-19 negative using the same PCR testing method. We denote this group as the *Challenge* cohort. The main reason to collect COVID-19 negative patient data from both the *Standard* and *Challenge* cohort is to specifically test for the ability to distinguish between COVID-19 and other respiratory illnesses (the use of the Challenge cohort), while collecting the necessary additional data to train a machine learning system (the use of the Standard cohort). Limitations in the data collection partnership prevented the collection of a larger cohort of patients who presented with respiratory illness but were COVID-19 negative.

PBS image data was collected using a clinically approved digital slide scanner (*Cellavision DM9600*, Lund, Sweden), which uses an oil immersion objective lens to capture multiple high resolution images per-patient centered upon stained (Wright-Giemsa) white blood cells (WBCs), with an average of 130 images captured per-patient. To preserve patient privacy, no additional data, such as demographic information, was collected. Fig 1 depicts the workflow of our automated analysis system.

### 2.2 Detecting COVID-19 from peripheral blood smears

After collecting high-resolution blood image data, our subsequent goal was to test the accuracy of a novel MIL algorithm to predict patient infection status. Unlike standard supervised machine learning problems that aim to establish a mapping between a single input image and a known output, this problem presents a somewhat unique challenge of accurately mapping a variable number of blood cell images to a single prediction of disease state. In a series of initial tests, we found that while predicting infection from a single image had poor performance (ROC-AUC of $\sim 0.7$), processing and averaging the predictions from multiple images of unique cells per patient dramatically improved algorithm accuracy. Based upon this key insight, we hypothesized that poor performance at the single image level stemmed from the fact that not every image has the requisite indicators to detect, or rule out, a COVID-19 infection, and that it would be beneficial to jointly optimize a cross-cell predictor aggregation strategy to increase diagnostic accuracy.

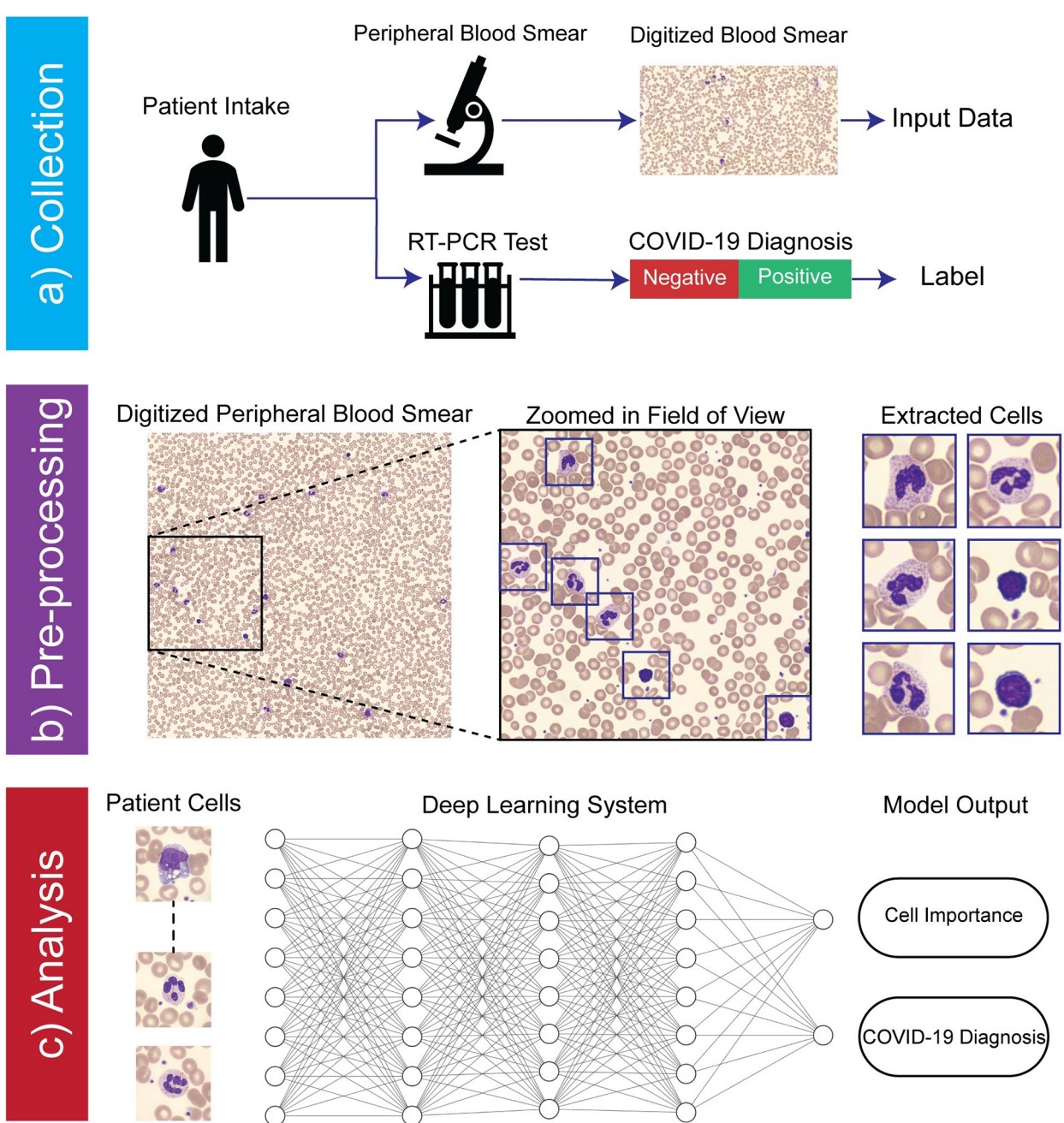

**Fig 1. Workflow of automated COVID-19 infection analysis from peripheral blood smear (PBS) image data. a)** Data collection procedure at Duke University Medical Center. Subjects were randomly selected from patient population subset who had both a COVID-19 RT-PCR test (ground truth labels) and images from a digital PBS scan (input data). All data was collected anonymously, retaining only the results of the RT-PCR test and digitized PBS images. **b)** Summary of pre-processing pipeline, where individual images of white blood cells are extracted from the full slide using a high-resolution oil-immersion microscope. **c)** Data analysis strategy. All cells from a patient are collectively analyzed by a deep neural network to produce both a COVID-19 diagnosis and per-cell importance score.

Accordingly, our final machine learning system used a hybrid of two different multiple instance learning (MIL) methods to map a patient blood imagery to a single diagnostic score (Fig 2). Each system branch offered unique theoretic benefits and in tandem formed an effective classifier that holistically examines patient PBS data. The figure encapsulates the various

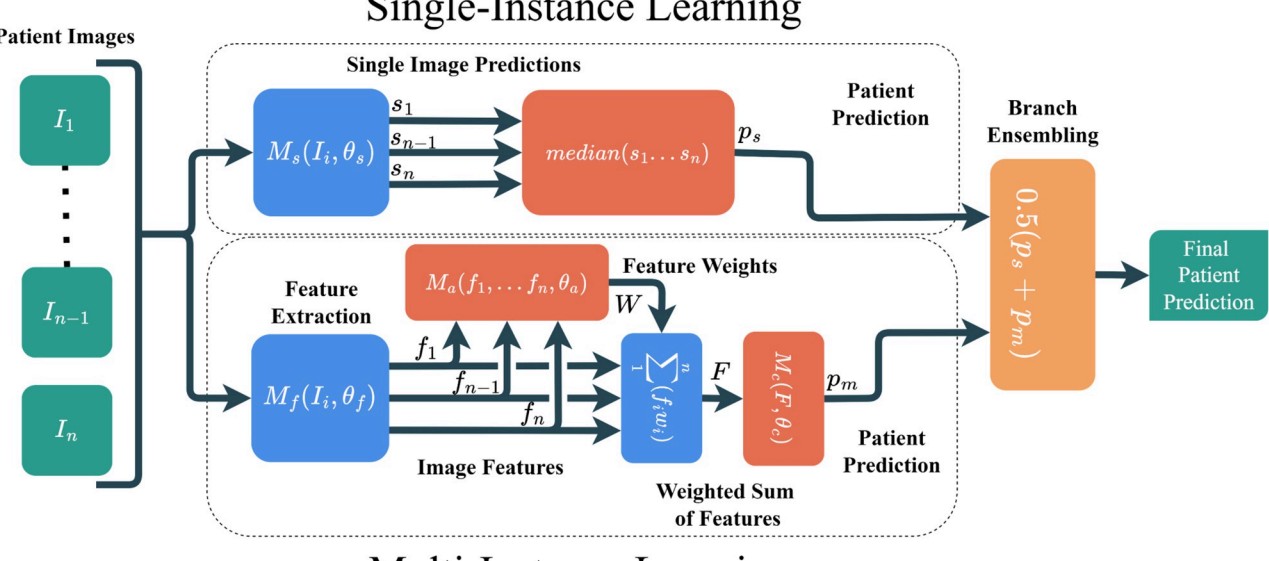

**Fig 2. Overview of hybrid machine learning system.** The Single Instance Learning (SIL) branch processes each image from a patient PBS scan individually. The outputs are then aggregated to produce a single prediction (using the median of the single image predictions). The MIL branch collectively analyzes all of a patient's images simultaneously, producing one prediction per patient. This is accomplished by first extracting learned per-image features, and then feeding those features into an attention module. The attention module assigns weights (summing to one) to each learned feature. The weights are used to compute a weighted sum across image features, the result of which is passed into a classification module to produce the MIL patient prediction. These two strategies are combined through ensembling, where the outputs of each branch are averaged to produce the final outcome.

discrete steps involved for each branch under distinct operators. Within the *Single-Image Learning* branch the main operator is the *Single Image Model* ($M_s$), which takes in a single patient image ($I_i$), and the model parameters ($\theta_s$) to produce single-image predictions. The median of these predictions is then fed into the ensembling step. For the *Multi-Image Learning* branch, we first apply a feature extraction model ($M_f$) to each patient image ($I_i$) using parameters $\theta_f$. Each of these feature vectors ($f_i$) is used by the attention mechanism ($M_a$) alongside the learned parameters for the attention mechanism, $\theta_a$, to produce a feature weight vector ($W$). Next, the feature vectors are weighted by elements of the feature weight vector and a weighted sum of the features is found ($F$). Finally, the patient prediction is made by applying a final learned model ($M_c$) to the weighted feature vector ($F$) with parameters $\theta_c$ to create the patient prediction for this branch.

Our final system allowed us to both effectively predict COVID-19 infection status, and identify which cells were most relevant to disease state prediction. We used a *Receiver Operator Characteristic* (ROC) curve to quantify the trade-off between diagnostic sensitivity and specificity of our new network, and obtained an area under the curve (ROC-AUC) of 0.90 and 0.89 on the standard and challenge cohorts respectively, shown in Fig 3. Additionally, we evaluated the accuracy of our predictions by assigning a pre-defined threshold (0.5 confidence score) to the outputs and calculating the percentage of subjects classified correctly. The accuracy of our network was 79% for subjects within the standard group, and 82% within the challenge group. Since the challenge cohort is composed of COVID-negative patients (but displaying severe acute respiratory illnesses), to evaluate the *challenge* cohort ROC-AUC, we randomly selected an equal number of COVID-19 positive patients from the *standard* group for comparison. We found that performance was roughly equivalent across both cohorts of data, suggesting that

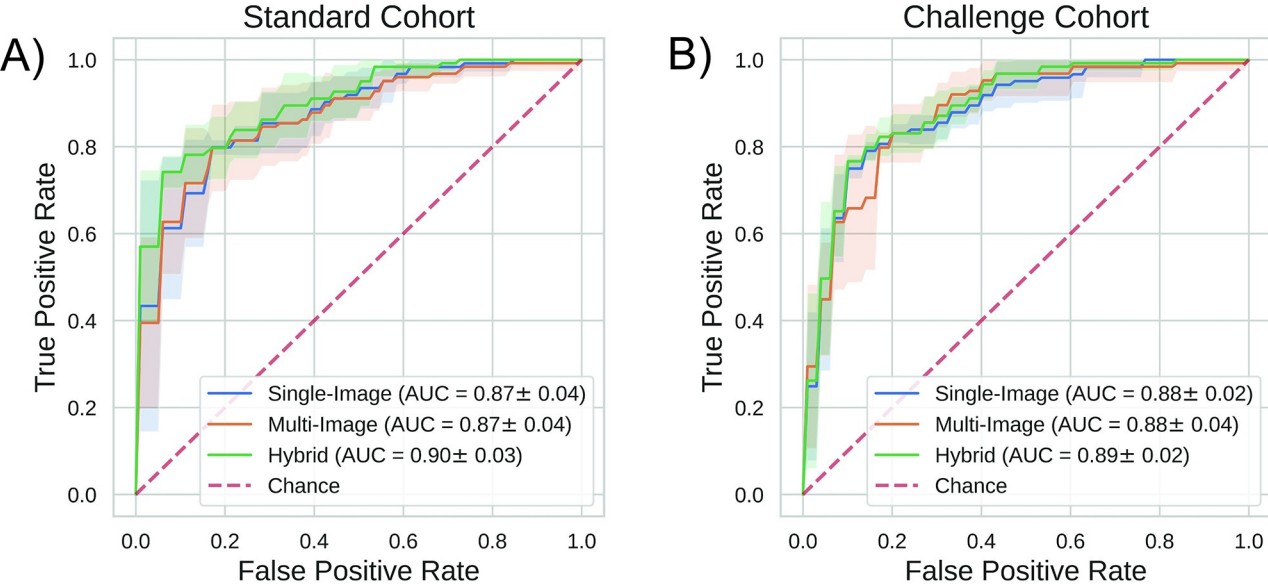

**Fig 3. Performance of COVID-19 diagnosis from blood cell morphology analysis as measured by the receiver operator characteristic (ROC).**
Classification accuracy was 79% and 82% for the standard and challenge cohorts respectively. A) Results reported are the average across the entire dataset, k-fold cross validation was used to maintain independence between the training and test sets. B) COVID-19 positive patients were randomly selected from the *standard* cohort to counterbalance the COVID-19 negative patients from the challenge cohort (ROC-AUC calculation requires both positive and negative examples).

the detection mechanism determined by our automated system is not influenced by hidden variables that may change as a function of scan time or patient cohort, but instead is correlated with the underlying disease.

## 2.3 Cell importance

In the process of predicting a patient's infection status, our system's MIL model uses an attention mechanism to generate a per-image importance score (trained jointly with the neural network). During a forward model pass, the importance score is used to create a weighted sum of the feature vectors from each image, which is then processed by a classifier module to generate a single diagnostic score per patient. Relative image importance scores may thus be directly examined during patient infection status inference, by translating each into a percentile score, where a 100th percentile cell image is the most important for the system to reach its COVID-19 diagnosis. With such an approach, we can jointly categorize cell images into their respective cell types (e.g., monocyte, basophil, etc.), and then compute statistical distributions of importance scores within and across cell types (Fig 4A).

From this analysis, we can draw several initial conclusions about the mechanisms used by our machine learning algorithm for COVID-19 detection. First, it is clear that neutrophils had a consistently higher importance value than other cell types. Second, cells classified as monocytes, platelets, and smudged cells (cells likely destroyed during slide preparation), had the lowest average scores and thus had less diagnostic use. Finally, the remaining cell types are moderately important, and likely contribute to an accurate classification, but less so than neutrophils (see example images in Fig 4B).

While these measures of importance alone are not enough to identify the exact mechanisms that our algorithm is using to perform diagnosis, they act as a rough guide to inform us which cells are more or less diagnostically relevant. Due to the inherently complex and non-linear

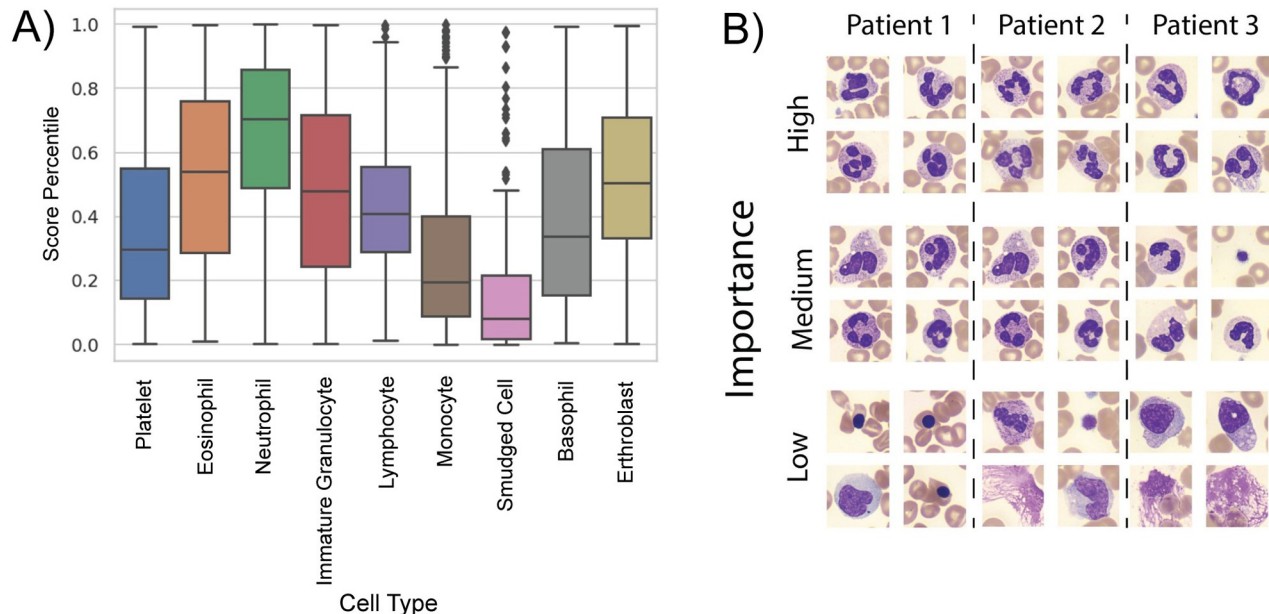

**Fig 4. Cell importance analysis.** A) Box plots of cell importance by detected white blood cell type. A higher score indicates higher importance. Neutrophils are the most consistently important cell type, whereas monocytes, smudged cells, and platelets are less important. The remaining cell types are moderately important and similar to each other. B) Examples of white blood cell images from three randomly sampled patients, split by importance level. Highly important cells are the cells of the patient with the highest score, medium importance cells were drawn from cells with scores closest to the 50th percentile of scores, and low importance cells were the four cells from each patient with the lowest scores.

nature of deep neural networks, it is difficult to identify precisely how classification decisions are made. However, our findings—that aspects of neutrophil morphology are important to identify a COVID-19 infection—are well supported by existing literature. Broad findings have recently connected COVID-19 to neutrophil-based abnormalities such as increased amounts of activated neutrophils in the bloodstream [20, 21] and elevated levels of neutrophil extracellular traps [22], among others [23].

To better understand potential cellular features which may be influencing how the system gauges the importance of individual cells we performed a visual analysis of a subset of the cells (100 in total) deemed "most important" by the system, for both the COVID-19 negative and positive cohorts. We looked for the presence of a discrete set of well known abnormalities affecting blood cells [24–26], these were: Dohle Bodies, Cytoplasmic Vacuoles, Hyposegmentation (either band formation or two discrete segments), Hypercondensation of Chromatin, Nuclear Bridges, Toxic Granulations, Nuclear Blebs, Abnormal Nucleus to Cytoplasm Ratio, Hypersegmentation, Inclusion Bodies, and Green Crystals. Each individual image of a cell was examined by a trained pathologist to determine which, if any, of the above abnormalities were present. We then compared the occurrence rate of abnormalities, by type, across the COVID-19 positive and negative cells. The results are shown in Fig 5.

Using logistic regression, we predicted the presence of each of these attributes within each cell from the COVID test status (negative vs. positive). We found that three of target attributes were present at a significantly higher rate for cells from COVID positive patients, they were Dohle Bodies ($\beta = 2.85$, $95\%CI[1.17, 5.77]$, $z = 2.68$, $p = 0.007$), Cytoplasmic Vacuoles ($\beta = 2.38$, $95\%CI[0.65, 5.31]$, $z = 2.21$, $p = 0.027$), and Hyposegmentation ($\beta = 1.29$, $95\%CI[0.46, 2.16]$, $z = 2.99$, $p = 0.003$). All three of these abnormalities are indicative of infection [24]. Therefore we can infer that our machine learning system may at least in part use these kinds of

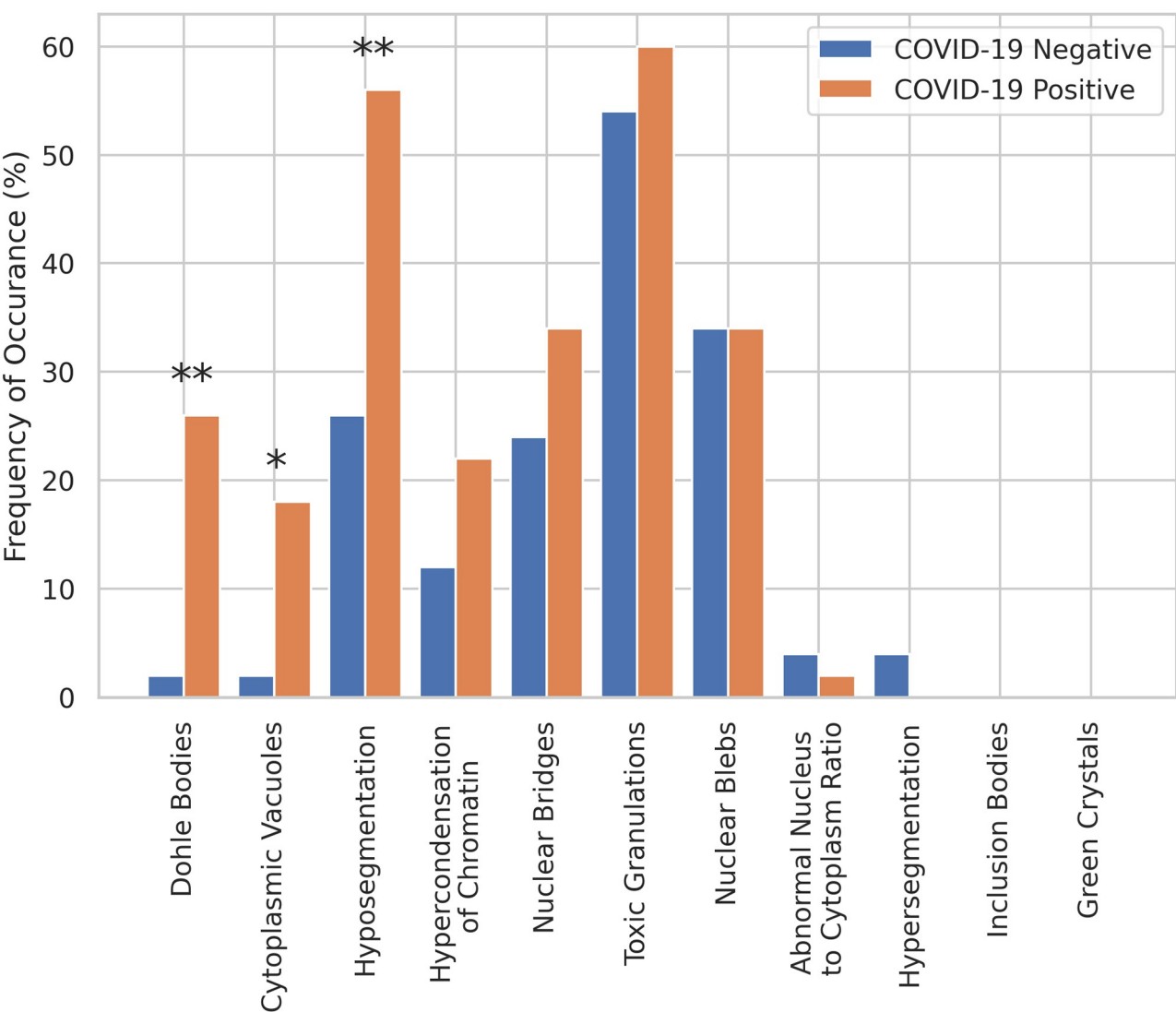

**Fig 5. The occurrence rate of abnormalities between COVID-19 negative and positive cells.** Significance is indicated on the three kinds of abnormalities that were found to significantly differ in frequency between conditions.

abnormalities as indicators for a COVID-19 infection. There was either no significant difference ($p >= 0.05$) or too few observations to make a statistical conclusion with our 100 cell sample for the remaining abnormalities.

## 2.4 Perturbation studies

To understand the spatial factors influencing our ability to detect COVID-19 from PBS images, we conducted a set of *Perturbation Studies*, where we manipulated aspects of the digital PBS image data in a controlled manner during neural network training and inference. In the interest of computational efficiency, these studies were only performed on a representative subset (three of the six folds used for k-fold cross-validation) of our data, and only for the *single-image* branch of our system. As noted above, PBS image data for each patient consists of 130 (on average) cropped images centered on a WBC, typically containing RBCs around the

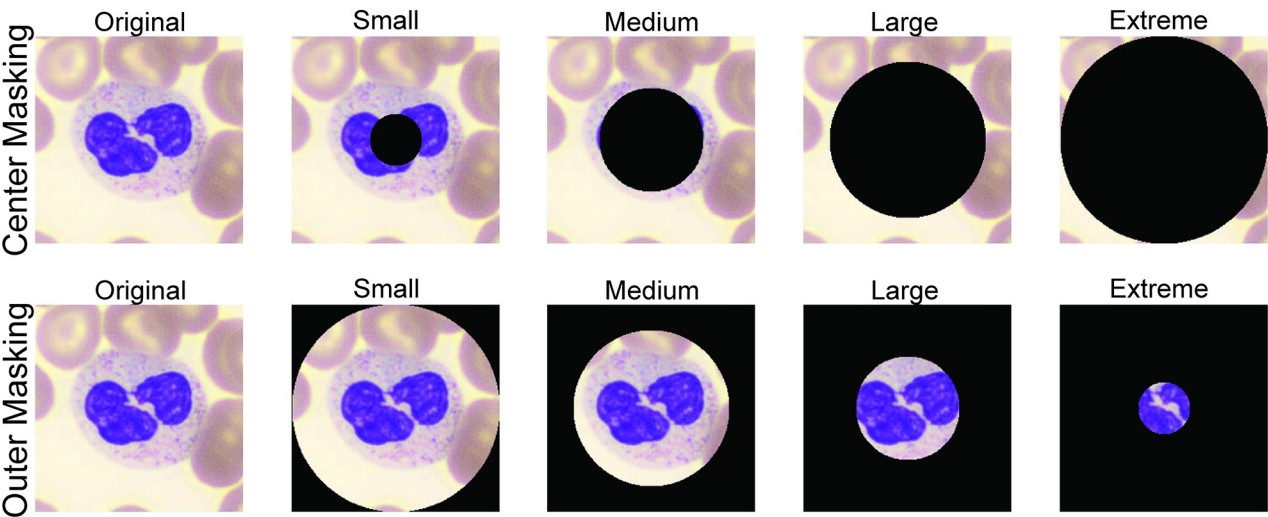

**Fig 6. Visual image perturbations applied during training.** Center masking occludes the white blood cell (which is always centered within the image). Outer masking occludes the red blood cells, which are in the area surrounding the white blood cell (towards the outside of the image).

periphery. Accordingly, we varied three unique aspects of these image datasets to help elucidate important factors for accurate COVID-19 diagnosis: the number of unique images per-patient, the amount of occlusion within the central image area that typically contains a white blood cell, and the amount of occlusion of the image periphery that contains red blood cells (see Fig 6). Occlusions were applied by zeroing pixel values in the same manner to all images within each patient's PBS image dataset.

While we can determine how the number of images per-patient influences performance simply by manipulating this value during inference, the latter two perturbations influence how the neural network processes images. Therefore, to effectively understand their impact, we retrained networks from scratch using occluded images to create a unique network per occlusion experiment. To jointly evaluate how the *quantity* of images per-patient influenced our ability to screen for COVID-19, we also varied the number of images available for COVID-19 diagnosis inference by randomly selecting up to $N$ images from each patient. This full process was repeated in three unique experiments with unmodified, center-occluded, and outer-occluded image data to produce the results summarized in Fig 7.

Across all configurations, we observe that a larger number of images per-patient leads to a higher quality screening result, with diminishing returns. This trend supports the notion that morphological indicators for COVID-19 infection are spread across many images (i.e., multiple blood cells). Somewhat surprisingly, with relatively few ($\sim 16$) images, the *original* configuration (no occlusions) reaches close to maximum performance, suggesting that while not every image has the requisite indicators to detect COVID-19, they are moderately prevalent within our dataset.

Examining the effect of the occlusions on the accuracy of our COVID-19 diagnosis predictions, we observe a negative relationship between occlusion size and prediction performance (Fig 7). Contrary to expectations, the system continues to perform fairly well (ROC-AUCs of $\sim 0.8$) under significant occlusion, if many images are used to make a prediction. These results point to several insights of interest. First, note that red blood cells are completely occluded for nearly all images in the extreme version of outer masking, and white blood cells are completely occluded for nearly all images in the extreme version of center masking.

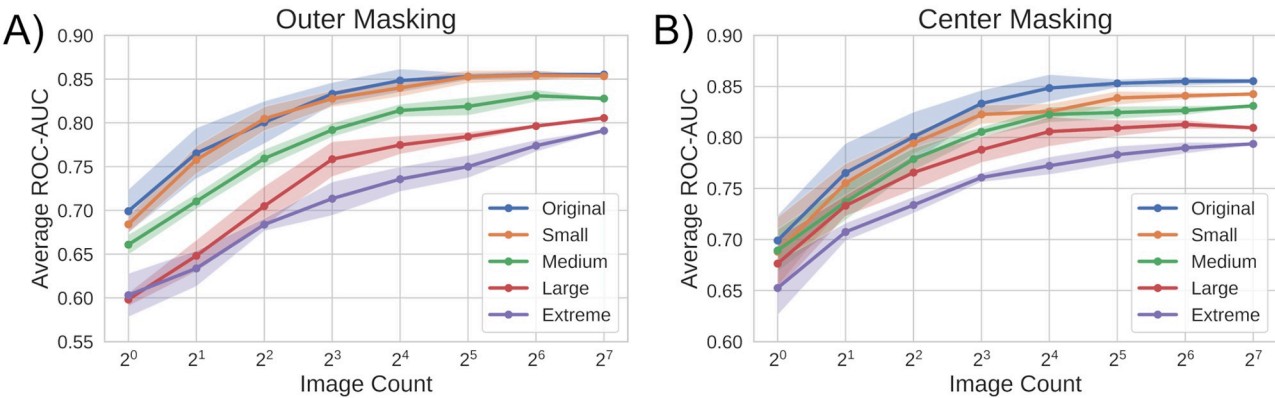

**Fig 7. Results of perturbation experiments.** Average ROC-AUC of screening predictions compared to the number of images (randomly selected across five trials) used to make the prediction.

Accordingly, it appears possible to at least weakly predict COVID-19 infection from either information about the white or red blood cells alone. Second, small "glimpses" of information that our model may see across hundreds of images per patient enable fairly accurate diagnosis. When only a few occluded images are available with data on a small number of cells, performance suffers greatly. Finally, while predictions *can* be made with only the red or white blood cell data, the best performance is consistently achieved when information about both cell types is jointly available.

These findings are reinforced by related preliminary evidence from the clinical domain. The impact of COVID-19 infection on white blood cell morphology has been observed in a number of studies [12, 13, 27, 28]. Our finding that red blood cells can morphologically change in response to COVID-19 infection is consistent with recent studies suggesting that red blood cell distribution width (RDW) is a significant predictor of illness [11], and that digital holographic videos of red blood cells can be used to assist with prediction of COVID-19 infection [29]. In conclusion, our new approach can jointly provide diagnostic predictions and lead to novel insights into how disease processes impact blood cell morphology. By aggregating trends across many hundreds of cells in a holistic manner, our DOBA pipeline offers a promising new direction for large-scale analysis of hematological and potentially alternative cytopathological image data in future tasks.

## 3 Methods

### 3.1 Data collection

We collected digital anonymized PBS images from patients at the Duke Medical Center (IRB Protocol 00105472). We preserved patient anonymity by only collecting PBS image data and COVID-19 infection status within the *standard* group of tested patients. The patients from the *challenge* group were selected by collecting PBS data from patients admitted to the medical intensive care unit with acute respiratory illness (from pneumonia or other acute respiratory failures) who tested negative for COVID-19. While used for cohort formation, this diagnostic information was not present during analysis. The SARS-CoV-2 infection test was performed with a Nasopharyngeal Swab based PCR test. All PBS were imaged with a *CellaVision DM9600* optical slide scanning system. The system captured high-resolution image segments (estimated 0.44$\mu m$ optical resolution) over a 360 × 360px image area, of which the inner 240 × 240 pixels

were used in our analysis. The images were typically centered on WBCs and almost always contained red blood cells. An average of 130 images were captured from each PBS.

The *standard* cohort included 236 patients, 125 of whom tested positive for COVID-19. Using k-fold cross-validation, provided through Scikit-Learn [30], ($k = 6$) we repeatedly split patients into training and test sets (equal proportion of COVID-19 positive patients within each set). There was no cross-over of patient data from set-to-set. Unless otherwise indicated, all performance metrics are reported as the average test-set performance across all six folds, where multiple independent models were trained from scratch exclusively for individual folds. This strategy enabled us to test our system on all available data while isolating the test data during the training process.

## 3.2 Data preprocessing and augmentation

We applied the same kind of data preprocessing and augmentations to all images used within this study. To prepare the data for processing by our neural networks we cropped and normalized the images. The cropping was done by taking the inner $240 \times 240$px from the original $360 \times 360$px image. We normalized the images by calculating a dataset wise mean and standard deviation for each color channel (red, blue, and green), then subtracting the mean and dividing by the standard deviation.

To augment our dataset we applied, in order, random horizontal and vertical flipping, random rotation, and color jittering. These augmentations were implemented using TorchVision [31] with parameters reported in Table 1. The aforementioned cropping step was done *after* the augmentations had been applied. During our perturbation experiments, we added a final masking step to our augmentation pipeline, replacing content within the mask area with zeros.

## 3.3 Machine learning system

Our machine learning system is a novel hybrid of two complementary multiple instance learning (MIL) approaches (shown in Fig 2). In the first branch, we used DenseNet-121 [32], a convolutional neural network (CNN), to process each image from a patient PBS image set independently. A single diagnostic label per patient was distributed across all of their individual PBS images. The CNN output a per-image confidence score, which was then combined across all images per patient by measuring the proportion of images classified as COVID-19 positive (0.5 threshold). This strategy may initially seem counter-intuitive, as not every image can be used to predict COVID-19 infection. If we consider these mislabelled images as a source

**Table 1. Data augmentation and pre-processing parameters (see Torchvision [31] for implementation details).**

| Parameter | Value |
|---|---|
| Random Horizontal Flipping | 50% probability |
| Random Vertical Flipping | 50% probability |
| Color Jitter | $\pm[0, 5]$% Brightness |
|  | $\pm[0, 5]$% Contrast |
|  | $\pm[0, 5]$% Saturation |
|  | $\pm[0, 5]$% Hue |
| Random Rotation | Uniformly sampled from $[-45, 45]$ |
| Center Crop | Center $240 \times 240$px |

Transformations are applied in the order they appear within the table. The same transformations were applied in both the single and multi-image models.

of label noise [33], then we can reconcile the effective performance of our technique despite this phenomenon with the fact that deep learning systems often succeed despite large amounts of label noise [34]. By training on individual images, the number of samples within our training dataset increases by several orders of magnitude to minimize overfitting and generalization issues, but at the expense of being unable to learn a method to integrate information *across* images.

To address this latter issue, we implemented a second MIL branch that adopted an attention based version of MIL, first shown by Ilse et al. [15]. In this strategy, a CNN extracts a feature vector from each patient's images and then applies an attention mechanism [35]. The attention mechanism used a small multilayer perceptron (MLP) with one hidden layer of size 64 and a *tanh* activation to produce a single *importance* value per image ($a_i$). Each importance value is placed into a softmax function to produce a set of feature weights $W$, such that $w_i = \frac{e^{a_i}}{\sum_{j=1}^{N} e^{a_j}}$

and $\sum_{j=1}^{N} w_j = 1$. These weights are used in conjunction with the extracted features to create a final per-image set feature vector. The combined feature vector is then propagated through a final MLP to produce a final patient-level classification. The entire pipeline is differentiable, so we trained this using only patient-wide labels.

For our single-image branch, we used the popular DenseNet-121 [32] CNN architecture, which has a $240 \times 240 \times 3$ input size. We optimized our model using the stochastic gradient descent optimizer included with PyTorch [36] paired with a cyclic learning rate schedule (following Smith [37]), linearly oscillating between high and low learning rates (1 and 0.0001 respectively) every 4000 gradient steps. The *BatchNorm* [38] layers within the DenseNet-121 architecture allowed for high learning rates (e.g. 1) without harming optimization. Empirically we found that when trained from random initialization, the *DenseNet-121* model performed better than its pre-trained alternatives.

In the multi-image branch of our system, we used the ResNet50 [39] CNN architecture to generate feature vectors and MLPs with one hidden layer to both generate attention scores and perform final classifications. While training with patient-level data reduces the number of unique training examples (one per-patient rather than per-image), it allows our model to learn relationships between per-patient cell images. However, as a single data point contains $N$ images instead of one within our system's second MIL branch, a forward pass of our model becomes $N$ times computationally larger. To accommodate this overhead, we adopted a simple modification: instead of inputting the entire set of images from each patient, we input a randomly selected subset of images. Based upon our image count findings (see Fig 7), we chose the size of this random subset to be 16. In addition to reducing memory requirements, we believe this sampling strategy acted as a regularization method to prevent the model from relying on relatively few images per patient for diagnosis formation.

Experimentally, we found that the multi-image branch was much more sensitive to learning rate, with large learning rates causing divergence. This may be because of how the gradient propagates (only a single label is used for 16 images), or the lack of BatchNorm in ResNet50. To accommodate this sensitivity, we modified our training strategy from the single image branch and used an adaptive optimizer (AdamW [40]) with a small, fixed, learning rate ($3e - 5$).

Model ensembling [16] was employed both within and between MIL branches. The output of each branch is the average of three independently trained models, and the output of our total system is the average of both branches. Ensembling substantially reduced average output error and could be additionally scaled up in the future for additional performance gains. Additionally, we applied data augmentation to individual blood cell images across both branches and their respective training regimes. The images were augmented using *Torchvision* [31] with standard horizontal/vertical flipping, random rotations, and color jittering. We tracked

performance on the held-out validation set during model training, with the final model used for evaluation being the iteration of the model which performed best on the validation set.

### 3.4 Evaluation of cell importance

The multi-image MIL branch assigned an importance score to each image (centered on a unique WBC) via its attention mechanism. While used collectively when predicting a patient's infection status, scores are functionally computed independently. To directly output per-image importance scores for additional analysis, we created a new sub-model, consisting of a subset of the trained multi-image branch (the feature extraction module and a portion of the attention module). All images across all patients were input into this sub-model and assigned importance scores before translating the distribution of values into percentiles ranging from *0th* (lowest) to *100th* (highest). To assess image importance as a function of cell type, we used a standard cell-type classifier to sort all images into one of nine standard categories: platelets, eosinophils, neutrophils, immature granulocytes, lymphocytes, monocytes, basophils, erythroblasts, and smudged cells (see additional details in S1 File. [4]).

### 3.5 Perturbation experiments

Spatial perturbations were introduced by modifying all images before being input into the neural network model. it was important to ensure perturbations were in place for the entirety of the training process (i.e., not just applied to test data, but also included during network training). Accordingly, all perturbation results are from independently trained models. We masked out a fixed number of pixels either from the center of the image within a given diameter, or the surrounding area outside of this central circle. For center masking, we used circles centered within the image with diameters of 60px, 120px, 180px, and 240px, for the minor, medium, major, and extreme configurations respectively. For the outer masking experiment, we followed the opposite approach, masking out *all but* the center of the image within the fixed diameters of 240px, 180px, 120px, and 60px for the minor, medium, major, and extreme configurations respectively.

Across all perturbation studies, we examined how the number of images being used per-patient influenced performance by reducing the number of images via random sub-selection. For example, if we wanted to test our system performance for a quantity of 16 images, we would randomly select 16 images from each patient to be included within the analysis, discarding all others. To ensure statistical significance with the random image sub-selection process, we repeated all analyses five times per model. Reported results include the average and standard deviation across all five trials.

### 3.6 Ethical approval

All experiments within this study were conducted with approval from the Duke University Health System Institutional Review Board (DUHS IRB). The DUHS IRB determined that the following protocol meets the criteria for a declaration of exemption from further IRB review as described in 45 CFR 46.101(b), 45 CFR 46.102 (f), or 45 CFR 46.102 (d), satisfies the Privacy Rule as described in 45 CFR 164.512(i), and satisfies Food and Drug Administration regulations as described in 21 CFR 56.104, where applicable.

### 3.7 Participant consent

A waiver was granted for this project because it was completed on archived sample data that was previously captured as part of standard care. Accordingly, the research involves no more

than minimal risk to subjects. The waiver of consent does not adversely affect the rights and welfare of the subjects, since utilized data was fully anonymized prior to their use minimal protected health information was utilized, and it was stored, kept secure and used following standard privacy and security practices at Duke Medical Center.

## 4 Discussion

In this paper, we present a MIL-based method to accurately diagnose the COVID-19 disease at a per-patient level from high-resolution morphological information across many blood cells and cell types. Besides the final aggregated decision, the proposed attention mechanism also provides cell-type importance, which can help pathologists to build valuable insights on which cell types are more diagnostically relevant. Moreover, by evaluating how different perturbations to our image dataset can affect the diagnosis results, we also studied which morphological features are more critical to the screening, opening a window into improving the explainability of machine learning approaches.

While our study shows promising outcomes, there are a few limitations of this pilot study. For example, the data collected from a single site for a short period of time. We hope that our new data-driven aggregation strategy can be applied for larger data collected from more diverse regions around the world, to serve as a starting point for future algorithmic development to understand the hematological impact of COVID-19 and other blood-related diseases.

The main limitation of this study is that the data collected represents a "snapshot" view of the patient's status. The peripheral blood smears were collected once, with no follow-up of additional measures afterwards. That means that we are unable to expand on our system's capabilities to begin to understand if we can determine the recovery trajectory of the patient, or predict the severity of the disease. Furthermore, we are limited to a peripheral blood smear per patient, so we cannot use our system to understand how the appearance of blood cells may change over time in response to an infection. In addition to limitations stemming from the method in which the samples were collected, there are potential limitations arising from the location of collection. In similar past work using data from a single institution, or medical device, may cause inherent bias in the data, making the method unable to perform effectively outside the environment it was developed in [41]. With the data currently available, we are unable to test the generalization capabilities of our system and therefore cannot gauge if it would be effective in alternate environments.

These limitations all lead towards potentially promising future work, where a more longitudinal and cross-institutional study of blood can be done, both by collecting more than one sample over time, and by collecting other measures in addition to peripheral blood smears (such as time to recovery, or measures of disease severity) to further understand the limits of our system's capabilities.

## Supporting information

**S1 File. Detailed description of model training and WBC classification.**
(PDF)

## Author Contributions

**Data curation:** Amey Chaware, Xing Yao, Xi Yang, Jadee Neff, Patricia Pittman, Chad McCall, Carolyn Glass.

**Formal analysis:** Colin L. Cooke, Kanghyun Kim, Shiqi Xu.

**Methodology:** Shiqi Xu.

**Resources:** Xing Yao.

**Supervision:** Chad McCall, Carolyn Glass, Xiaoyin Sara Jiang, Roarke Horstmeyer.

**Writing – original draft:** Colin L. Cooke.

**Writing – review & editing:** Xing Yao.

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
