## [Decision Letter · Decision Letter 0]

7 Apr 2022

PDIG-D-22-00017

Deep Optical Analysis of Peripheral Blood Smear Images for Next Generation Screening

PLOS Digital Health

Dear Dr. Xu,

Thank you for submitting your manuscript to PLOS Digital Health. After careful consideration, we feel that it has merit but does not fully meet PLOS Digital Health's publication criteria as it currently stands. Therefore, we invite you to submit a revised version of the manuscript that addresses the points raised during the review process.

We look forward to receiving your revised manuscript.

Kind regards,

Dukyong Yoon

Academic Editor

PLOS Digital Health

Journal Requirements:

1. Your co-authors, Colin L. Cooke (colin.cooke@duke.edu), Amey Chaware (amey.chaware@duke.edu), Xi Yang (xi.yang@duke.edu), Patricia Pittman (patricia.pittman@duke.edu), Chad McCall (chad.mccall@duke.edu), Xiaoyin Sara Jiang (jiang009@duke.edu), and Roarke Horstmeyer (rwh4@duke.edu), have not confirmed authorship of the manuscript. We have resent them the authorship confirmation email; however please check that the above email address for them is correct and follow up personally to ensure they confirm. Please note that we cannot pass your manuscript to Production until we have received confirmations from all co-authors.

2. Please amend your detailed Financial Disclosure statement. This is published with the article, therefore should be completed in full sentences and contain the exact wording you wish to be published.

State what role the funders took in the study. If the funders had no role in your study, please state: “The funders had no role in study design, data collection and analysis, decision to publish, or preparation of the manuscript.”

3. Please update the completed 'Competing Interests' statement. Please declare all competing interests beginning with the statement “I have read the journal's policy and the authors of this manuscript have the following competing interests:”.

4. We ask that a manuscript source file is provided at Revision. Please upload your manuscript file as a .doc, .docx, .rtf or .tex. If you are providing a .tex file, please upload it under the item type ‘LaTeX Source File’ and leave your .pdf version as the item type ‘Manuscript’.

5. Please provide separate figure files in .tif or .eps format and remove any figures embedded in your manuscript file. Please also ensure that all files are under our size limit of 20MB. If you are using LaTeX, you do not need to remove embedded figures.

For more information about how to convert your figure files please see our guidelines: https://journals.plos.org/digitalhealth/s/figures

6. We have noticed that you have uploaded supporting information but you have not included a list of legends. Please add a full list of legends for all supporting information files (including figures, table and data files) after the references list.

Additional Editor Comments (if provided):

Reviewers' comments:

Reviewer's Responses to Questions

**Comments to the Author**

1. Does this manuscript meet PLOS Digital Health’s publication criteria? Is the manuscript technically sound, and do the data support the conclusions? The manuscript must describe methodologically and ethically rigorous research with conclusions that are appropriately drawn based on the data presented.

Reviewer #1: Yes

2. Has the statistical analysis been performed appropriately and rigorously?

Reviewer #1: Yes

Reviewer #2: Yes

3. Have the authors made all data underlying the findings in their manuscript fully available (please refer to the Data Availability Statement at the start of the manuscript PDF file)?

Reviewer #1: No

Reviewer #2: Yes

4. Is the manuscript presented in an intelligible fashion and written in standard English?

Reviewer #1: Yes

Reviewer #2: Yes

5. Review Comments to the Author

Reviewer #1: The manuscript is interesting in its exploration to build a deep learning model to predict Covid from PBS. It is well written and describes the methods rigorously often lacking in other ML manuscripts. The study design is interesting especially the pertubation studies which investigates the cell type of relevance. 

Having said this, there are certain shortcomings of this study which need to be addressed to answer some key questions and to make such a model utilizable in the clinical setting. 

1. It is strongly suggested to the authors to reword the manuscript title. It currently makes no mention of what exactly was investigated in this study let alone give any indication that the use case was Covid. Secondly, the phrase "next generation screening" is also confusing as no mention of using the model as a screening test is made in the manuscript. NGS also has strong word association with the genomics domain and can be confusing for readers and clinicians who might have different expectations going in when they read this manuscript.

2. Study cohort selection- Section 2.1 describes the standard and challenge cohorts and the patient populations from whom these smears were used. It is unclear why the rest of the 47% standard cohort was included if only 53% pts were RT-PCR positive. Moreover what is the difference then between the 47% standard cohort and the challenge cohort who too tested RT-PCR negative.

3. It is strongly recommended to include the data preprocessing section from supplemental to the main manuscript before Section 3.2. Understanding this is key in addition to the model parameters and architecture.

4. Was this model validated on an external PBS dataset acquired outside of the primary medical center mentioned in this study? Is it a possibility to perform it? If not, please mention this as a limitation in discussion. Evidence for generalizability is commonly expected from ML studies. 

5. The biggest critique of this study is its inability to explain what cell features does the ensemble model recognize and weigh heavily in its inference. Correlation at the cell type level is very generic and do not indicate why it would be unique to a Covid infection. E.g. Neutrophils which are most highly associated with the model, increase commonly across a variety of bacterial and viral infections. The authors are strongly recommended to provide a posthoc explanation of what features the model relies on to make its decision e.g. inclusion bodies, changes nucleocytoplasmic ratio etc

6. An understanding of how long such cell specific Covid associations and morphological changes can be identified is desirable. For instance, what happens when a patient recovers from Covid and has a followup blood exam a month after. Will this model still infer the PBS to be Covid positive if the cell changes haven't returned to baseline? It is suggested to examine model performance on longitudinal data if possible. This would really add strong clinically actionable insight to the current study.

Reviewer #2: This study suggest a multi-instance learning-based approach to aggregate high-resolution morphological information across many blood cells and cell types. This can be used as a part of clinical decision support system. 

Overall descriptions and figures about the workflow of the proposed system are well expressed. In particular, it explains the meaning of the results in detail. This enhances the persuasion of the proposed system. By analyzing the importance of each cell type, cells with higher and lower infection were identified. Also, this study indirectly elucidated the mechanism by which the model detects COVID-19 through perturbation studies. It breaks the conventional perception that the mechanism of ML models are unclear and the universal reliability is low.

Minor commnet:

1. The abbreviation of SIL is not indicated. Also, in the figure 2, if there is no problem changing the 'branch' term to 'learning', I recommend that you change it. 

2. A sentence often extends over 3 to 4 lines, making it difficult to grasp the context. In many cases, the results and reasons are expressed in one sentence, and I recommend you to divide the sentences.

3. Method section: Please describe in more detail how the attention mechanism was applied to the model. 

Section 2.2: The description in Figure 2 refers to the attention mechanism, but it is difficult to understand in Figure itself. Please modify the figure more intuitively. Also, there is not enough description of the formulas in the figure (ex. Ma, Mc...).

6. PLOS authors have the option to publish the peer review history of their article (what does this mean?). If published, this will include your full peer review and any attached files.

**Do you want your identity to be public for this peer review?** For information about this choice, including consent withdrawal, please see our Privacy Policy.

Reviewer #1: No

Reviewer #2: No

**Comments to the Author**

1. Does this manuscript meet PLOS Digital Health’s publication criteria? Is the manuscript technically sound, and do the data support the conclusions? The manuscript must describe methodologically and ethically rigorous research with conclusions that are appropriately drawn based on the data presented.

Reviewer #2: Yes

---

## [Decision Letter · Decision Letter 1]

21 Jun 2022

A multiple instance learning approach for detecting COVID-19 in peripheral blood smears

PDIG-D-22-00017R1

Dear Mr Xu,

We are pleased to inform you that your manuscript 'A multiple instance learning approach for detecting COVID-19 in peripheral blood smears' has been provisionally accepted for publication in PLOS Digital Health.

Best regards,

Dukyong Yoon

Academic Editor

PLOS Digital Health

Reviewer Comments (if any, and for reference):

Reviewer's Responses to Questions

**Comments to the Author**

1. If the authors have adequately addressed your comments raised in a previous round of review and you feel that this manuscript is now acceptable for publication, you may indicate that here to bypass the “Comments to the Author” section, enter your conflict of interest statement in the “Confidential to Editor” section, and submit your "Accept" recommendation.

Reviewer #1: All comments have been addressed

Reviewer #2: All comments have been addressed

2. Does this manuscript meet PLOS Digital Health’s publication criteria? Is the manuscript technically sound, and do the data support the conclusions? The manuscript must describe methodologically and ethically rigorous research with conclusions that are appropriately drawn based on the data presented.

Reviewer #1: Yes

Reviewer #2: Yes

3. Has the statistical analysis been performed appropriately and rigorously?

Reviewer #1: Yes

Reviewer #2: Yes

4. Have the authors made all data underlying the findings in their manuscript fully available (please refer to the Data Availability Statement at the start of the manuscript PDF file)?

Reviewer #1: Yes

Reviewer #2: Yes

5. Is the manuscript presented in an intelligible fashion and written in standard English?

Reviewer #1: Yes

Reviewer #2: Yes

6. Review Comments to the Author

Reviewer #1: Kudos to the authors for performing posthoc analysis with the pathologist based on my comments. My questions and suggestions have been satisfactorily addressed. I believe this will be an interesting manuscript for the readers as well.

Reviewer #2: (No Response)

7. PLOS authors have the option to publish the peer review history of their article (what does this mean?). If published, this will include your full peer review and any attached files.

**Do you want your identity to be public for this peer review?** For information about this choice, including consent withdrawal, please see our Privacy Policy.

Reviewer #1: No

Reviewer #2: None
